# Photobiomodulation Therapy through a Novel Flat-Top Hand-Piece Prototype Improves Tissue Regeneration in Amphioxus (*Branchiostoma lanceolatum*): Proposal of an Ethical Model for Preclinical Screening

Matteo Bozzo [1], Claudio Pasquale [2], Francesco Cuccaro [1], Sara Ferrando [1], Angelina Zekiy [3], Simona Candiani [1,†] and Andrea Amaroli [3,*,†]

1   Department of Earth, Environment and Life Sciences, University of Genoa, 16132 Genoa, Italy; matteo.bozzo@edu.unige.it (M.B.); francescocuccaro91@gmail.com (F.C.); sara.ferrando@unige.it (S.F.); candiani@unige.it (S.C.)
2   Department of Surgical and Diagnostic Sciences, University of Genoa, 16132 Genoa, Italy; clodent@gmail.com
3   Department of Orthopedic Dentistry, Faculty of Dentistry, First Moscow State Medical University (Sechenov University), 119991 Moscow, Russia; zekiy82@bk.ru
*   Correspondence: andrea.amaroli.71@gmail.com
†   These authors contributed equally to this work.

**Abstract:** Despite the literature providing compelling evidence for the medical efficacy of photobiomodulation (PBM) therapy, its consistency in terms of accuracy and standardization needs improving. Identification of new technology and reliable and ethical biological models is, therefore, a challenge for researchers working on PBM. We tested the reliability of PBM irradiation through a novel delivery probe with a flat-top beam profile on the regenerating amphioxus *Branchiostoma lanceolatum*. The caudalmost $9 \pm 1$ myotomes, posterior to the anus, were excised using a sterile lancet. Animals were randomly split into three experimental groups. In the control group, the beam area was bounded with the 635-nm red-light pointer (negligible power, <0.5 mW) and the laser device was coded to irradiate 810 nm and 0 W. In Group laser-1, the beam area was bounded with the same 635-nm red-light pointer and irradiated at 810 nm, 1 W in CW for 60 s, spot-size 1 cm$^2$, 1 W/cm$^2$, 60 J/cm$^2$, and 60 J; irradiation was performed every day for two weeks. In Group laser-2, the beam area was bounded with the same 635-nm red-light pointer and irradiated at 810 nm, 1 W in CW for 60 s, spot-size 1 cm$^2$, 1 W/cm$^2$, 60 J/cm$^2$, and 60 J; irradiation was performed on alternate days for four weeks. We observed that PBM improved the natural wound-healing and regeneration process. The effect was particularly evident for the notochord. Daily irradiation better supported the regenerative process.

**Keywords:** low-level laser therapy; light therapy; phototherapy; regeneration; wound healing; ethical model; 3Rs principle

## 1. Introduction

The chemical and physical environment of the primitive Earth is enforced to explain the origin of life and its evolution [1]. All life forms need energy for existence, but only photosynthetic organisms adopted light-energy conversion and elected some wavelength ranges to survive in the biosphere. Conversely, the animal cell did not choose light as a source of energy for its metabolism [2].

However, during the last 50 years, growing evidence has suggested that visible and near-infrared light wavelengths can modulate energy metabolism also in non-photosynthetic cells [3]. Indeed, animal-cell photoacceptors can interact with photons, absorb their photoenergy, and, through a photosynthetic-like process, transform the radiant energy into chemical energy [4]. Just like for the chloroplast in a plant, the key role in that conversion

process is played by the mitochondrion, either directly through the chromophores of respiratory chain complexes [4–7] or indirectly after modulation of calcium homeostasis [8], nitric oxide release [9], and vibrational energy of water [10], lipid [11], and transforming growth factor-beta (TGF-β) [12]. This role of the mitochondrion can be explained through the theory of endosymbiont models and the parallel and convergent evolution of the mitochondrion and chloroplast from ancestral bacteria [13]. From a clinical point of view, the modulation of the cell energy metabolism improves tissue dysfunction during nerve, bone, endothelial, muscle, and epithelial regeneration [3] as well as in mental disorders [14]. Furthermore, effects on neurotransmitter release have been demonstrated [15]. The medical treatment exploiting this process is known as photobiomodulation (PBM), which was preferred to the term low-level laser therapy (LLLT). Basically, photobiomodulation works because visible and (near-)infrared light can modulate cell metabolism and homeostasis without causing significant thermal increases.

Recently, Amaroli et al. [8] discussed the effect of PBM on cellular pathways that commonly govern life/death processes and, according to Hamblin [16], suggested that all life forms could be responsive to energization by photons. However, PBM within the animal kingdom was only partially investigated, and information is particularly lacking for invertebrates, which can potentially represent suitable bioethical models for pre-clinical research. Additionally, although the literature provides compelling evidence for the medical efficacy of PBM therapy [3], its consistency in terms of accuracy and standardization needs improving [17]. Limits in the delivery instruments and the standardization of related therapies may indeed dramatically affect the applications of PBM. The identification of new technologies and reliable and ethical biological models is, therefore, a challenge for PBM researchers. Furthermore, fractional calculus [18] and electromagnetic numerical models [19] could support the description of photo-induced effects and their reproducibility. On this basis, we tested the reliability of PBM therapy through a novel delivery probe [17] on the European amphioxus *Brachiostoma lanceolatum* Pallas, 1778.

Amphioxus, or cephalochordates, are benthic, marine filter-feeding invertebrates that, together with tunicates and vertebrates, constitute the chordate phylum. Its morphological and genomic features and phylogenetic position as the earliest branching chordate (Figure 1) make amphioxus the best model organism to study the evolutionary changes that occurred during chordate evolution. The body plan of amphioxus is indeed that of a prototypical chordate: a dorsal, hollow nerve cord with an anterior brain, an axial notochord, segmented muscles, pharyngeal gill slits, and an endostyle. Moreover, many organs and structures of amphioxus are homologs to those of vertebrates [20]. From the operative point of view, the maintenance of amphioxus in captivity is relatively easy, and alternative reliable husbandry methods have been established in different laboratories worldwide, with or without direct access to seawater [21]. Pharmacological treatments are easily performed, as the animals can adsorb drugs present in the seawater through their skin and gut, especially during development [22], and bead implants have been successfully employed to locally administer chemicals to adult tissues [23]. Gene editing is also feasible, although not as easily as in other organisms [24]. In sum, amphioxus is not only interesting per se, but it can also provide a better proxy of vertebrates than any other invertebrate model, including flies and nematodes. Since a vertebrate-like epimorphosis process orchestrated by canonical Wnt/bone morphogenetic protein (BMP)-signaling pathways is involved in amphioxus tail regeneration [23,25,26], this organism could be proposed as a pre-clinical animal model of regeneration, which meets the principles of the 3Rs (Replacement, Reduction, and Refinement) [27].

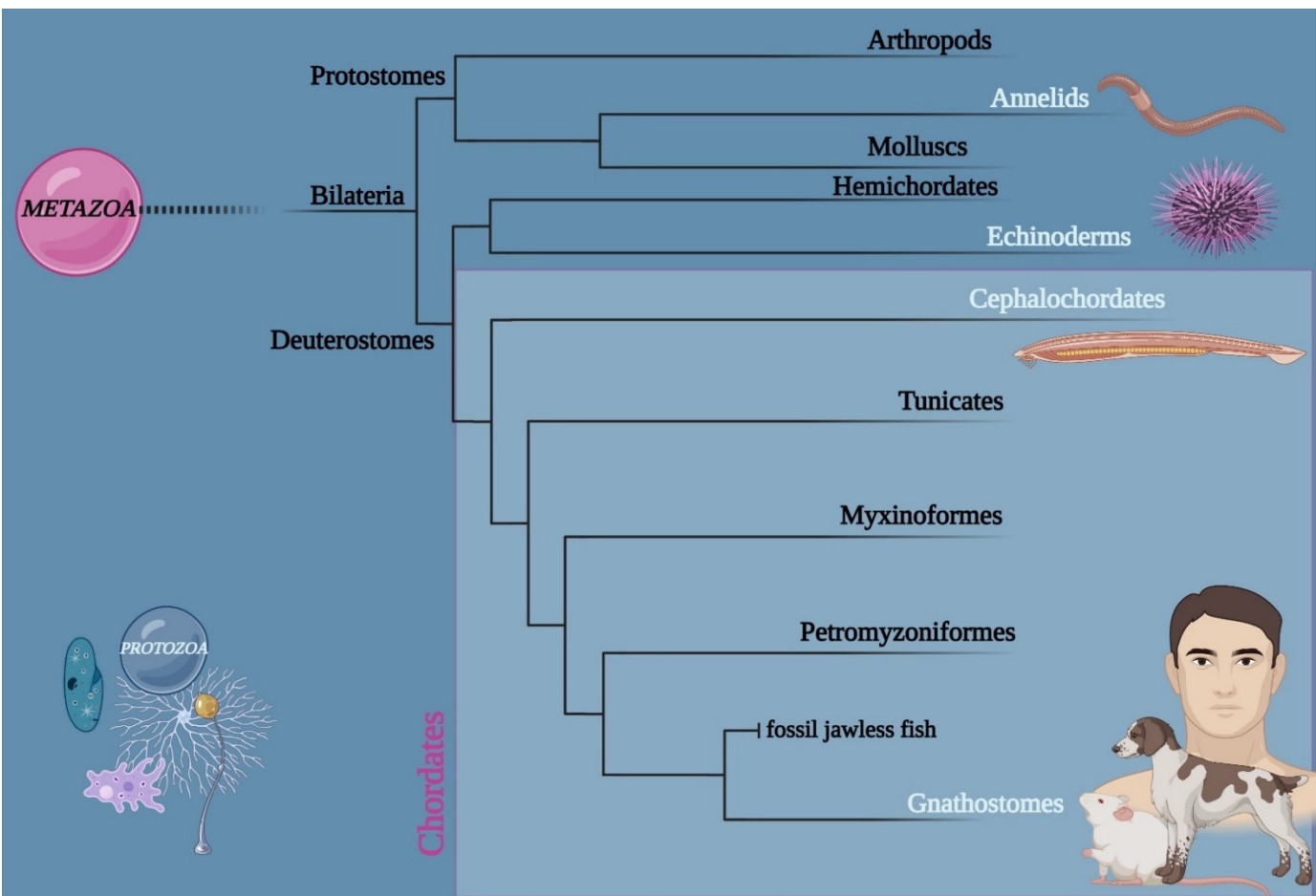

**Figure 1.** Phylogenetic position of Cephalochordates (amphioxus) among the Metazoa. Taxa written in white contain species that are affected by exposition to photobiomodulation with 810 nm, 1 W, and 60 J/cm$^2$, in continuous-wave for 60 s on a spot-size area of 1 cm$^2$ (1 W/cm$^2$; 60 J). *Paramecium primaurelia* (Protozoa) [28]; *Dictyostelium discoideum* (Protozoa) [29]; *Dendrobaena veneta* (Annelida) [30]; *Paracentrotus lividus* (Echinodermata) [31]; *Mus musculus* (Gnathostomata, Mammalia) [32]; *Homo sapiens* (Gnathostomata, Mammalia) [33]. Created with BioRender.com.

Therefore, the predictor variable of our research was both the employment of a novel hand-piece with a flat-top (FT-HP) beam profile to deliver the PBM therapy on the animal model amphioxus and the regenerative ability of *B. lanceolatum*. The primary endpoint was the improvement of tissue regeneration. The secondary endpoint was detecting any adverse effects. Animal specimens were irradiated through the higher power and fluence therapy of 810 nm, 1 W, and 60 J/cm$^2$, in a continuous wave (CW) for 60 s on a spot-size area of 1 cm$^2$ (1 W/cm$^2$; 60 J). The laser therapy was chosen according to our previous works on the FT-HP characterization of isolated mammalian mitochondria [17] and organisms [28–33] (Figure 1).

## 2. Materials and Methods

### 2.1. Animal Model: Branchiostoma lanceolatum

Adult *Branchiostoma lanceolatum* specimens were collected at Argelès-sur-Mer (France) [34] in April, before the beginning of the spawning season, and maintained in a seawater facility under standard conditions [22]. Experiments started in November of the same year, after around seven months of acclimatization to captivity.

### 2.2. Device for Irradiation of Photobiomodulation

To improve the knowledge of PBM therapy and its consistency, an 810 nm diode laser (GaAlAs) device (Garda Laser, 7024 Negrar, Verona, Italy) was used. Such a device was equipped with our novel hand-piece prototype, the FT-HP, which can irradiate through a flat-top beam profile [17].

According to our recent study on the characterization of FT-HP on mitochondria isolated from mammals [17], the PBM therapy was administered at the power of 1 W in CW for an exposure time of 60 s and on a spot size of 1 cm$^2$, which allowed for generating a power density of 1 W/cm$^2$ and a fluence of 60 J/cm$^2$ (energy administered = 60 J) (Figure 2). The accuracy of the laser parameter irradiated was secured by the Pronto-250 power meter (Gentec Electro-Optics, Inc. G2E Quebec City, QC, Canada). To avoid beam reflection, the Petri dishes with the animal sample were posed on an absorbing-material mat.

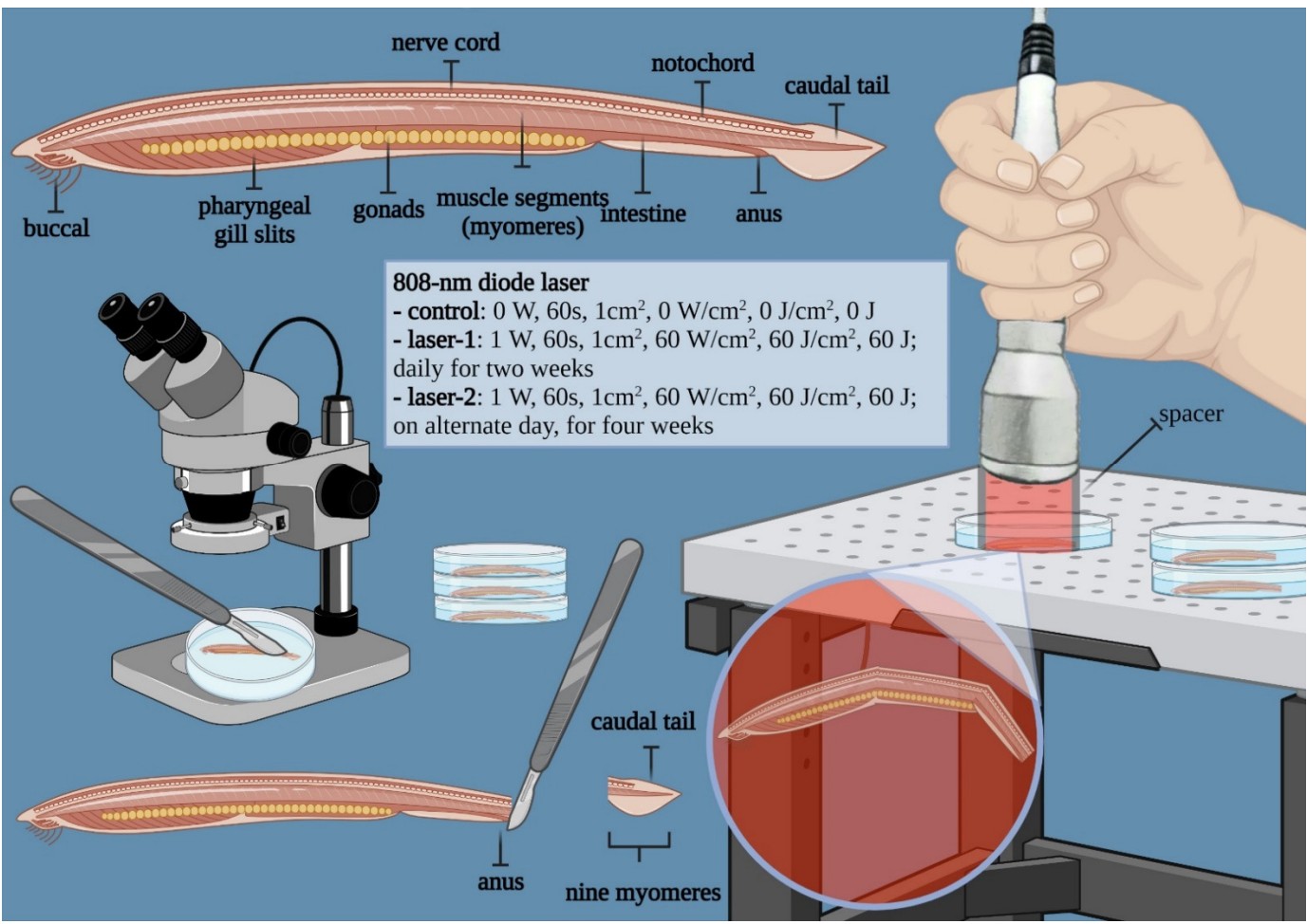

**Figure 2.** Experimental setup. Top left: anatomical description of amphioxus. Excision, irradiation of the specimens, and 810-nm diode-laser parameters were used. Created with BioRender.com.

Adverse events due to a possible undesirable thermal effect were avoided by monitoring the irradiation with a thermal camera FLIR ONE Pro-iOS (FLIR Systems, Inc. designs, Portland, OR, USA) (dynamic range: −20 °C/+400 °C; resolution 0.1 °C).

### 2.3. Experimental Setup

*Branchiostoma lanceolatum* specimens were accustomed to growing in Petri dishes (100 mm × 20 mm) filled with sterile, filtered seawater and kept at the temperature of 19 °C. During the process of adaptation and for the whole experiment, the animals were fed once every two days with *Tysocrisis* sp., while the water was changed daily. In total,

36 specimens of *B. lanceolatum* of the same size (total body length = 4 ± 0.2 cm) were selected for the experimental procedure. The experimental design is briefly described in Figure 2. Animals were anesthetized with MS222 (250 mg/mL in seawater), and the caudalmost 9 ± 1 myotomes were excised using a sterile lancet [25]. The cut result was from the posterior to the anus.

Animals were randomly split into three experimental groups, which received different treatments. Control group: the beam area was bounded with the 635-nm red-light pointer (negligible power, <0.5 mW), and the laser device was set to irradiate 810 nm, 0 W for 60 s, spot-size 1 $cm^2$, 0 $W/cm^2$, 0 $J/cm^2$, and 0 J. Group laser-1: the beam area was bounded with the 635-nm red-light pointer (negligible power, <0.5 mW), and the laser device was set to irradiate 810 nm, 1 W in CW for 60 s, spot-size 1 $cm^2$, 1 $W/cm^2$, 60 $J/cm^2$, and 60 J; irradiation was performed every day for two weeks. Group laser-2: the beam area was bounded with the 635-nm red-light pointer (negligible power, <0.5 mW), and the laser device was set to irradiate 810 nm, 1 W in CW for 60 s, spot-size 1 $cm^2$, 1 $W/cm^2$, 60 $J/cm^2$, and 60 J; irradiation was performed on alternate days for four weeks. In all groups, the angle of incidence of the laser beam was orthogonal to the surface on which the animals were lying.

Despite the HP-FT being able to keep the power constant from contact to many centimetres [17], a spacer of 2 cm was employed (Figure 2) to maintain the irradiation distance constant and to keep animals inside the irradiated-spot area. To avoid bias, the splitting of the groups, irradiation, and specimen analysis were performed by different operators, and the Petri dishes were maintained in a blinded manner. The animals were monitored every day, and the specimens were analyzed as described below.

### 2.4. Gross Morphology and Histology

The posterior end of the amputated animals was monitored daily with a Leica MZ APO stereo microscope (Leica, Wetzlar, Germany) equipped with a Moticam 10+ camera (Seneco S.r.l., Milan, Italy). Representative specimens were selected for histological analysis at different time points. Briefly, animals were euthanized by an MS222 overdose, fixed with 10% neutral buffered formalin at 4 °C for 24 h, dehydrated, and embedded in paraplast (P3558, Sigma-Aldrich Corporation, Saint Louis, MO, USA). Then, 5 μm-thick longitudinal sections were produced using a Leica RM2125 RTS microtome (Leica, Wetzlar, Germany) and stained with Masson's trichrome stain (04-010802, Bio-Optica, Peschiera Borromeo, Italy). Slides were analyzed and photographed using a Leica DMRB microscope (Leica, Wetzlar, Germany) equipped with a Moticam 3+ camera (Seneco S.r.l., Milan, Italy).

### 3. Results

### 3.1. Morphological Analysis

Morphological analysis (Figure 3) showed the natural ability of amphioxus to regenerate the amputated posterior part of its body. Importantly, all samples were able to produce a new tail anlage by week 8, and abnormal morphology was not observed. However, a faster regeneration was observed in Group laser-1, subjected to daily irradiation (Figure 3A–H′). Indeed, an organized blastema-like structure was observed in these samples several days earlier than in both the controls and the samples irradiated on alternate days (Group laser-2). Additionally, longer and better-organized tails appeared in animals from Group laser-1 compared to those from Group laser-2 and the control group. Of note, the regenerative process in animals from Group laser-2 follows a temporal progression comparable to that observed in the control animals.

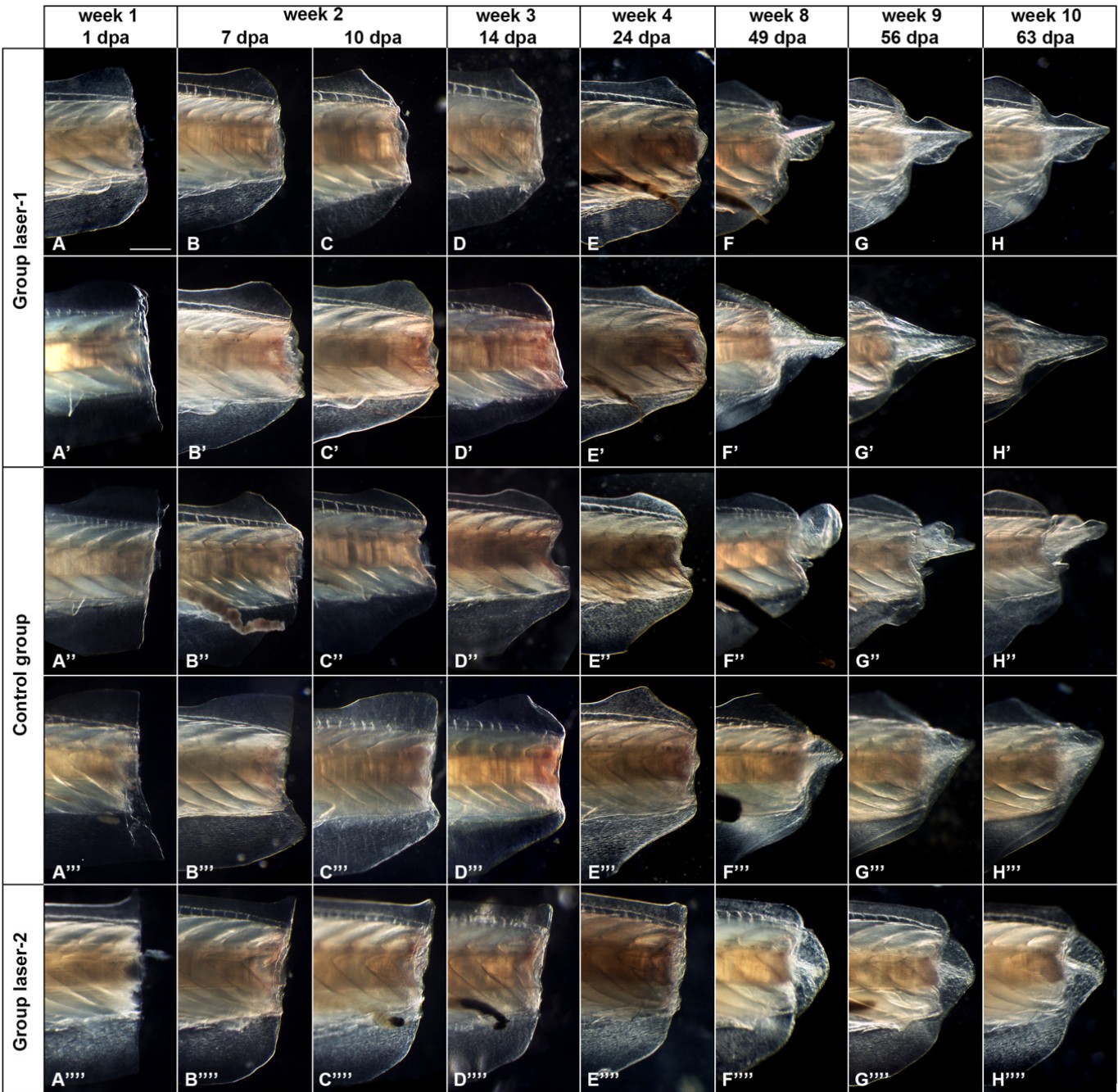

**Figure 3.** Gross morphology of the regenerating tail over the course of 10 weeks. (**A–H,A′–H′**) show two representative specimens from Group laser-1. (**A″–H″,A‴–H‴**) show two representative specimens from the control group. (**A⁗–H⁗**) show a representative specimen from Group laser-2. The scale bar in (**A**) is 0.5 mm and is valid for all panels.

### 3.2. Histological Analysis

Histological analyses performed 24 days post-amputation (dpa) revealed a similar picture in all specimens assayed (Figure 4A–C). The wound was sealed with a simple epithelium. At the posterior end of the nerve cord, neural cells are reorganized to form a cavity (terminal ampulla). The terminal part of the notochord lost its typical stack-of-coin structure and was constituted by loosely organized mesenchymal cells. The notochordal collagenous sheath was interrupted at the posterior end, so the mesenchymal cells of the notochord were in contact with the mesenchymal cells of the blastema (asterisks in Figure 4). The muscle fibers of the posterior-most myotomes also appeared unorganized (arrows in Figure 4).

Similar analyses conducted at a later time point (63 days post-amputation) highlighted considerable differences between the treatment groups (Figure 4A′–C′). In the control group, the cells of the terminal myotomes and notochord (arrowhead in Figure 4B′) still appeared poorly organized, which was similar to what was observed at 24 days post-amputation. Conversely, both laser-irradiated groups had terminal myotomes with well-formed fibers and stack-of-coin cells up to the posterior end of the notochord (Figure 4A′,C′). Overall, the histological organization of both groups is comparable to that of the normal unamputated tail. The main difference between treatment groups appeared to be the amount of new tissue produced, with the specimens from Group laser-1 having a significantly longer newly formed tail (Figure 4A′,C′).

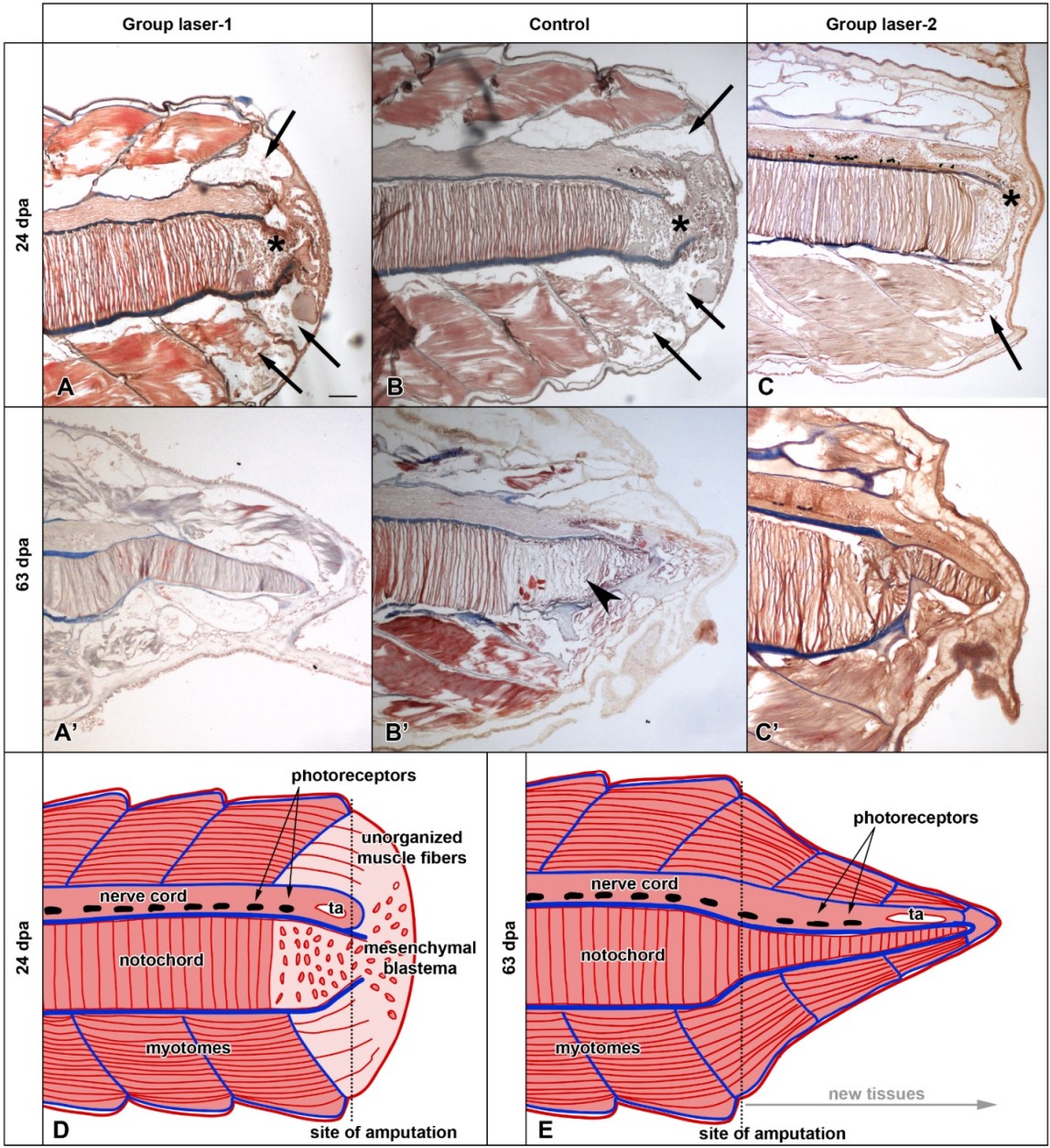

**Figure 4.** Histology of the regenerating tail 24 and 63 days post-amputation (dpa) of representative specimens from Group laser-1 (**A**,**A′**), control (**B**,**B′**), and Group laser-2 (**C**,**C′**). (**D**,**E**): Scheme of the regenerative process as seen in Group laser-1. ta, terminal ampulla; asterisks indicate mesenchymal cells; arrows indicate muscle fibers; arrowhead indicates the notochord. The scale bar in (**A**) is 100 μm and valid panels (**A**–**C′**).

### 3.3. Thermal Monitoring

No substantial thermal increase (T increment < 2 °C) was detected during the 60-s irradiation by monitoring with the thermal camera FLIR ONE Pro-iOS.

## 4. Discussion

Our results showed the regenerative ability of amphioxus, which followed the spatial and temporal organizations described by Somorjai and colleagues [25]. This supports the consistency of our experimental setup and the choice of the samples, which meet the experimental requirement for laser irradiation. Complying with our primary variable, the laser therapy of 810 nm, 1 W, 60 s, and 60 J/cm$^2$ in CW mode irradiated through a novel flat-top beam profiled hand-piece improved the natural wound-healing and regeneration process of *Branchiostoma lanceolatum*. Indeed, regenerated structures in laser-irradiated animals presented better tissue organization compared to the control group. The effect was particularly evident for the notochord. Additionally, the posology of therapy administration significantly affected the tail-regeneration process. Indeed, if both daily and every-other-day irradiation supported better histological organization, the continuous frequency of administration significantly sped up the process. A similar outcome was observed in *Dendrobaena veneta* (Annelida), where the daily irradiation of the same 810 nm PBM therapy allowed a more equilibrated employment of cell energy, by mitigating the effect on inflammation and tissue degeneration and likely affecting the energy metabolism of the cells [30].

Somorjai and collaborators [25] suggested that local progenitors resembling the stem cells involved in vertebrate tail regeneration are activated during amphioxus tail regeneration, and, like those, their ability to regenerate drastically reduces with aging. Actually, in vertebrates, it is well established that the proliferative ability and plasticity of stem cells decline with aging. Stem cells isolated from aged individuals, indeed, show a weaker expansion potential, rendering their autologous transplantation a challenge [35]. However, as recently reviewed by Amaroli and collaborators [36], photons can modulate the agenda of vertebrate stem cells through the improvement of energetic metabolism and anti-inflammatory and osteogenic capacity. On the other hand, PBM not only improves stem cell viability and proliferation but, in accordance with precise parameters, also prompts these multipotent cells towards a predefined lineage commitment and secretome. In particular, the authors pointed out that 810-nm therapy, the same that we irradiated on amphioxus, was able to improve cell respiration and metabolic energy production [6] as well as the osteogenic agenda of murine mesenchymal stromal cells [32,37].

Somorjai and collaborators [25] found that β-catenin is specifically involved in the nascent notochord in regenerating amphioxus, suggesting an implication of the canonical Wnt pathway during tail regeneration. Expression of the BMP signaling in undifferentiated cells of the early bud-stage tail blastema was further described. In addition, phosphohistone H3-positive nuclei, indicative of cell proliferation, were found to be differentially located in the blastema of young amphioxus compared to that of older individuals [38]. Of note, Agas and collaborators [39] recently demonstrated that PBM modulates the cellular agenda of undifferentiated murine osteoblast-precursors through activation of both Wnt and SMAD 2/3-β-catenin pathways. They suggested that PBM, after a primary target interaction with the cytochromes of the mitochondrial respiratory chain, can modulate cellular cascades such as Wnt, TGF, and BMP, which enhance cell differentiation and proliferation. Similarly, murine stem cells were affected to increase the synthesis of TGF-β1, down-regulate pro-inflammatory cytokines [interleukin (IL)-6, and IL-17], and up-regulate anti-inflammatory ones (IL-1rα and IL-10) [32,37]. An increment of the osteogenic runt-box-related transcription factor (Runx2) and osterix production was also observed. The process was supported through cytoskeleton reorganization toward the upregulation of key proteins involved in actin nucleation, such as neuronal Wiskott–Aldrich syndrome protein (N-WASP), actin-related protein (Arp2/3) (specifically the p34/ArpC2 subunit), and cortactin [32].

Additionally, Martins and co-workers showed that PBM drives massive epigenetic histone H3 modifications, stem cell mobilization, and accelerated epithelial healing [40]. Therefore, the faster and improved regeneration process that we observed in amphioxus could be the consequence of the increased metabolic energy and proliferative and differentiative cellular-pathways activation. This seem mimic cell-rejuvenating-like process induced by laser light and described in stem cells from aged patients [36].

## 5. Conclusions

The prompt and predictable responses of amphioxus to PBM and the absence of adverse effects support our primary and secondary variables. Indeed, the consistency of the regeneration process observed after the PBM-therapy irradiation corresponds with our previous data on both cellular and organismal responses to the same laser parameters [9,30–33]. Particularly, the coherence between the regulative and restorative effects observed in amphioxus and those of murine models [32] and humans [33] endorses the possible employment of amphioxus as an ethical, preclinical screening model for PBM therapies. From an evolutionary point of view, the 810 nm, 1 W, 60 s, and 60 J/cm$^2$ in CW-mode PBM therapy can positively affect the cell or tissues homeostasis in a wide range of eukaryotes, such as Protozoa [29], Annelids [30], Echinoderms [31], Vertebrates [32], and Cephalochordates (Figure 1). Basically, PBM through near-infrared light exploits the ability of certain molecules of being both photoacceptive and capable of regulating conserved cellular pathways. Therefore, regeneration, as well as other restorative effects of PBM, appears to be a byproduct, which was conserved in evolution.

**Author Contributions:** Conceptualization, M.B., S.C., S.F. and A.A.; methodology, M.B., S.C. and A.A.; software, M.B. and A.A.; validation, S.C. and A.A.; formal analysis, M.B. and A.A.; investigation, M.B., F.C., C.P., S.C. and A.A.; resources, A.Z.; data curation, M.B., S.C. and A.A.; writing—original draft preparation, M.B., S.C. and A.A.; writing—review and editing, M.B., S.C., S.F. and A.A.; supervision, S.C.; funding acquisition, S.C. All authors have read and agreed to the published version of the manuscript.

**Funding:** This study was supported by the University of Genoa (Italy) through FRA (Fondi per la Ricerca di Ateneo) grants to S.C.

**Institutional Review Board Statement:** Not applicable.

**Data Availability Statement:** The datasets generated and/or analyzed during the current study are available from the corresponding author upon reasonable request.

**Acknowledgments:** We would like to thank Alessandro Bosco, for the outstanding work done and the support for the realization of the flat-top hand-piece prototype previously characterized in [17], and Mirco Raffetto, for their help during in the peer-review process. Moreover, the authors are indebted to Hector Escrivá (Observatoire Océanologique de Banyuls-sur-Mer, France) for providing *Branchiostoma lanceolatum* adults.

**Conflicts of Interest:** The authors declare no conflict of interest.

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
