# Peer review of "Photobiomodulation Therapy through a Novel Flat-Top Hand-Piece Prototype Improves Tissue Regeneration in Amphioxus (Branchiostoma lanceolatum): Proposal of an Ethical Model for Preclinical Screening"

_photonics, doi:10.3390/photonics9070503_

Round 1

Reviewer 1 Report

The manuscript submitted by Bozzo, Amaroli, and co-workers described the reliability of PBM irradiation through a novel delivery probe with a flat-top beam profile on the regenerating amphioxus Branchiostoma lanceolatum. The authors claimed to observe that PBM improved the natural wound healing and regeneration process. They also found the effect was particularly evident for the notochord and daily irradiation supported efficient regeneration process. The manuscript is well written and the conclusions presented in this manuscript are supported by experimental data. This reviewer feels that the work is potentially interesting and this manuscript can be accepted for publication in Photonics in its current form.

Author Response

Dear Reviewer 1,

Thank you so much for appreciating our paper and supporting it

The main proofreading was highlighted in green. 

Best Regards

Andrea Amaroli and co-workers

Reviewer 2 Report

The manuscript ID photonics-1812580 mainly presents a short communication related to photobiomodulation therapy assisted by a particular flat-top hand-piece prototype. Results describing an improvement in tissue regeneration in amphioxus are shown. Please see below a list of comments to the authors:

1.   How was monitored the incident irradiation and absorbed irradiation in the samples?

2.   How much light is transmitted by the samples?

3.   Is there an important reflection or scattering in the samples that importantly decreases the incident energy in the samples studied?

4.   Is there an influence of the incident polarization in the photobiomodulation therapy?

5.   How was selected the wavelength and angle of incidence of the beam for the experiments?

6.   Inhomogeneous effects with flat top beam profile can be obtained in photobiomodulation therapies. Fractional models have been proposed to describe the photoenergy transfer in biological samples. The authors are invited to discuss about potential implications and perspectives in advanced laser therapy. You can see for instance: https://doi.org/10.1016/j.ijthermalsci.2022.107734

7.   Is there a motivation to use coherent or incoherent light for the main findings? Incoherent light seems to provide different results in photobiomodulation therapy, you can see for instance: https://doi.org/10.3389/fmedt.2022.871196

8.   How much light is transmitted by the samples? The authors wrote “To avoid beam reflection the Petri dishes with the animal sample were posed on an absorbing material mat” It is required to analyze if the heat provided by the material mat absorbing the transmitted light can be responsible of a contribution in the results.

9.   In my opinion, some of the collective citations should be split in order to justify the references by clear individual expressions.

10.               A proofreading is suggested. In the abstract is written: “the laser device was coded to irradiate 810-nm and 0W.” I wonder if it means that the sample was in darkness.

Author Response

Dear Reviewer 2,

First of all, thank you so much for appreciating our paper and for the precious advice that improved our work.

Please find in the journal’s template our revisions highlighted in green. 

Point-by-point answers to your comments are listed below.

Q1: How was monitored the incident irradiation and absorbed irradiation in the samples?

Q2: How much light is transmitted by the samples?

Q3: Is there an important reflection or scattering in the samples that importantly decreases the incident energy in the samples studied?

Q4: Is there an influence of the incident polarization in the photobiomodulation therapy?

A1,2,3,4: Thank you for your questions. The data are shown according to the style of photobiomodulation papers. I know that the data presentation and the experimental set-up accuracy can appear poor if compared to studies of physic sciences or electromagnetic fields engineering. However, to date, the background cannot support deeper descriptions. This is the main problem in the photobiomodulation topic that in my opinion affect the reproducibility of the effects from in vitro to in vivo. In two recent and previous reviews, we showed that the photobiomodulation process is supported by the intrinsic proprieties of molecules involved in the abiotic life origins and that are now involved in the physiological process of animal and human cells (more generally, no-plant cells) [1], as well as the obstacles that must be overcome to improve photobiomodulation reliability [2].

For this reason, we started a collaboration with Prof. Bruno Bianco (professor emeritus for its studies on bioelectromagnetism) and the full professor Mirco Raffetto at Genoa University.

Our first paper was recently published [3] and we are ready to submit a second work correlated to that. Unfortunately, the data are at the moment not sufficient to clarify your questions.

Additionally, to improve the standardisation and the reliability of the photobiomodulation our teams designed the hand-piece used in our paper to deliver the therapy. The characterisation of the flat-top handpiece was described in our recent work [4]. The more homogeneous irradiation delivered through the flat-top design improved indeed the therapy and reduced many variables correlated to the different amounts of energy delivered on the spot size area with respect to the employment of standard hand-piece or fibres.

We are working and I hope the photobiomodulation researchers will work to improve photobiomodulation through specific measures and predictive mathematical model design supporting the next studies.

[1] Amaroli A, Ravera S, Zekiy A, Benedicenti S, Pasquale C. A Narrative Review on Oral and Periodontal Bacteria Microbiota Photobiomodulation, through Visible and Near-Infrared Light: From the Origins to Modern Therapies. Int J Mol Sci. 2022 Jan 25;23(3):1372. doi: 10.3390/ijms23031372. PMID: 35163296; PMCID: PMC8836253.

[2] Amaroli A, Pasquale C, Zekiy A, Benedicenti S, Marchegiani A, Sabbieti MG, Agas D. Steering the multipotent mesenchymal cells towards an anti-inflammatory and osteogenic bias via photobiomodulation therapy: How to kill two birds with one stone. J Tissue Eng. 2022 Jul 5;13:20417314221110192. doi: 10.1177/20417314221110192. PMID: 35832724; PMCID: PMC9272199.

[3] Amaroli A, Benedicenti S, Bianco B, Bosco A, Clemente Vargas MR, Hanna R, Kalarickel Ramakrishnan P, Raffetto M, Ravera S. Electromagnetic Dosimetry for Isolated Mitochondria Exposed to Near-Infrared Continuous-Wave Illumination in Photobiomodulation Experiments. Bioelectromagnetics. 2021 Jul;42(5):384-397. doi: 10.1002/bem.22342. Epub 2021 May 18. PMID: 34004023; PMCID: PMC8251649.

[4] Amaroli A, Arany P, Pasquale C, Benedicenti S, Bosco A, Ravera S. Improving Consistency of Photobiomodulation Therapy: A Novel Flat-Top Beam Hand-Piece versus Standard Gaussian Probes on Mitochondrial Activity. Int J Mol Sci. 2021 Jul 21;22(15):7788. doi: 10.3390/ijms22157788. PMID: 34360559; PMCID: PMC8346075.

Q5:   How was selected the wavelength and angle of incidence of the beam for the experiments?

A5: We used the orthogonal incidence because is in accordance with the main experimental set-up in photobiomodulation studies and therefore, simplifies the experimental approach and comparison with literature. It makes data repeatability of the tests and it is similar to the in vivo and clinical application generally performed in contact mode with the probe perpendicular to the irradiated area. We added a sentence in section 2.3 specifying the angle of incidence.

As specified at the end of introduction section and in figure 1 “The laser therapy was chosen according to our previous works on the FT-HP characterization of isolated mammalian mitochondria [17] and organisms [28-33] (Figure 1).”

Q6:   Inhomogeneous effects with flat top beam profile can be obtained in photobiomodulation therapies. Fractional models have been proposed to describe the photoenergy transfer in biological samples. The authors are invited to discuss about potential implications and perspectives in advanced laser therapy. You can see for instance: https://doi.org/10.1016/j.ijthermalsci.2022.107734

A6: On living organisms, there is no doubt that an exogenous electromagnetic field can alter the heat transfer mechanism in the stressed organism. This lies in the properties of the visible and infrared light wavelengths and the power/energy of the therapy irradiated. For this reason, the employment of lasers in medicine covers many areas from surgery to photobleaching, photodynamic therapy and photobiomodulation, as examples.

Different fields with specific needs can be supported by wide laser skills. Photobiomodulation works because visible and (near-)infrared light can modulate cell metabolism and homeostasis without causing significant thermal increases. Therefore, the range of wavelengths in the visible and near-infrared range, as well as the therapies parameters, must not induce a thermal increase. As pointed out in our review [1], wavelengths show different interactions with the cell photoacceptors. For photobiomodulation to occour, laser parameters have to be chosen to avoid thermal increase. For this reason, a thermo camera was employed to monitor the experiments. [pag. 4 ln 125: thermal camera FLIR ONE Pro-iOS (FLIR Systems, Inc. designs, 97070 Portland, Ore-gon, U.S.A.) (dynamic range: -20°C/+400°C; resolution 0.1°C).]. Of course, this avoids macroscopic thermal increase and we have no information at the micro- and nano-scopic level. However, as clearly described in the text, the therapy employed in our work has been previously characterised on extracted mitochondria (the main target of 810-nm), unicellular model organisms, invertebrate model of regeneration and developmental model, without detecting detrimental effects. Also, clinical studies demonstrated the efficacy of the therapy employed in our work without adverse effects at 6 months follow-up. 

The laboratory laser used in the work https://doi.org/10.1016/j.ijthermalsci.2022.107734 has different skills and interactions with the cell and tissues with respect to the 810 nm laser, as well as the primary variables of the works are very different.

However, we appreciated your suggestion that can be advised for future works in terms of improving the photobiomodulation reliability and reproducibility. We added a sentence in the introduction. We also added a sentence to clarify that "photobiomodulation works because visible and (near-)infrared light can modulate cell metabolism and homeostasis without causing significant thermal increases"

Q7:   Is there a motivation to use coherent or incoherent light for the main findings? Incoherent light seems to provide different results in photobiomodulation therapy, you can see for instance: https://doi.org/10.3389/fmedt.2022.871196

A7: There are discussions in photobiomodulation between coherent or incoherent, continuous wave or pulsed mode and combined wavelengths or a wavelength alone.

As previously discussed in this cover letter the main problem of photobiomodulation is the reliability and the standardization of the results from in vitro to in vivo (animals and patients). There are many variable that can influence the photobiomodulation final results and that can modulate a positive or adverse effect [2].

We worked to reduce the variables. And because coherent light well supported our results we choose it, as well as we choose a continuous wave, flat-top irradiation and a wavelength alone.

In our condition, we observed in our previous studies an 96% increment in ATP production.

Indeed, coherent light makes PBM analysis easier as there is no ambiguity on the spectrum of electromagnetic stress that is used; to fully define an experiment that uses an inconsistent source, the spectrum of the radiation provided by the source should also be indicated.

Q8:   How much light is transmitted by the samples? The authors wrote “To avoid beam reflection the Petri dishes with the animal sample were posed on an absorbing material mat” It is required to analyze if the heat provided by the material mat absorbing the transmitted light can be responsible of a contribution in the results.

A8: Yes indeed, but as described in the text in photobiomodulation we have to avoid consistent thermal increase and therefore, the irradiation was monitored through a thermal camera FLIR ONE Pro-iOS (FLIR Systems, Inc. designs, 97070 Portland, Ore-gon, U.S.A.) (dynamic range: -20°C/+400°C; resolution 0.1°C) [pag. 4 ln 125]. In accordance with our previous studies [4,5], no consistent thermal increase was observed (<2°C).

We added a sentence in the result section.

[5] Ravera S, Bertola N, Pasquale C, Bruno S, Benedicenti S, Ferrando S, Zekiy A, Arany P, Amaroli A. 808-nm Photobiomodulation Affects the Viability of a Head and Neck Squamous Carcinoma Cellular Model, Acting on Energy Metabolism and Oxidative Stress Production. Biomedicines. 2021 Nov 18;9(11):1717. doi: 10.3390/biomedicines9111717. PMID: 34829946; PMCID: PMC8615884.

Q9:   In my opinion, some of the collective citations should be split in order to justify the references by clear individual expressions.

A9: We subdivided references where necessary.

  1. A proofreading is suggested. In the abstract is written: “the laser device was coded to irradiate 810-nm and 0W.” I wonder if it means that the sample was in darkness.

A10: Yes, it indicates that the control animals were treated exactly like the experimental groups (for instance they were kept out of water for a few minutes) but the laser was switched off so that they were not irradiated with 810-nm light.

Main proofreading was highlited in green. 

Hoping to encounter your appreciation, we send you our

Best Regards

Andrea Amaroli and co-workers

Round 2

Reviewer 2 Report

I agree with the importance of these results as a communication paper and with the reviewed version of the manuscript. The results can be relevant and the presentation of this photobiomodulation therapy assisted by a particular flat-top hand-piece prototype can be useful for future research. Then I can recommend this work for publication in present form.